# Expert consensus on the important chronic non-specific neck pain motor control and segmental exercise and dosage variables: An international e-Delphi study

Jonathan Price[1,2]*, Alison Rushton[2,3]◦, Vasileios Tyros[4], Nicola R. Heneghan[2]◦

**1** Musculoskeletal Physiotherapy Services, Birmingham Community Healthcare NHS Foundation Trust, Birmingham, United Kingdom, **2** College of Life and Environmental Sciences, Centre of Precision Rehabilitation for Spinal Pain (CPR Spine) School of Sport, Exercise and Rehabilitation Sciences, University of Birmingham, Birmingham, United Kingdom, **3** School of Physical Therapy, Western University, Ontario, Canada, **4** Edgbaston Physiotherapy Clinic, Birmingham, United Kingdom

◦ These authors contributed equally to this work.
* Jonathan.price2@nhs.net

## Abstract

### Background

Chronic non-specific neck pain is highly prevalent, resulting in significant disability. Despite exercise being a mainstay treatment, guidance on optimal exercise and dosage variables is lacking. Combining submaximal effort deep cervical muscles exercise (motor control) and superficial cervical muscles exercise (segmental) reduces chronic non-specific neck pain, but evaluation of optimal exercise and dosage variables is prevented by clinical heterogeneity.

### Objective

To gain consensus on important motor control and segmental exercise and dosage variables for chronic non-specific neck pain.

### Methods

An international 3-round e-Delphi study, was conducted with experts in neck pain management (academic and clinical). In round 1, exercise and dosage variables were obtained from expert opinion and clinical trial data, then analysed thematically (two independent researchers) to develop themes and statements. In rounds 2 and 3, participants rated their agreement with statements (1–5 Likert scale). Statement consensus was evaluated using progressively increased *a priori* criteria using descriptive statistics.

### Results

Thirty-seven experts participated (10 countries). Twenty-nine responded to round 1 (79%), 26 round 2 (70%) and 24 round 3 (65%). Round 1 generated 79 statements outlining the interacting components of exercise prescription. Following rounds 2 and 3, consensus was achieved for 46 important components of exercise and dosage prescription across 5 themes

**Data Availability Statement:** All relevant data are within the manuscript and its Supporting Information files.

**Funding:** This work was completed as part of a Clinical Health Research MRes at the University of Birmingham, UK funded by Health Education England and National Institute for Health Research (HEE/ NIHR ICA Programme Pre-doctoral Clinical Academic Fellowship, Mr Jonathan Price, ICAPCAF-2018-01-117). The views expressed in this publication are those of the author(s) and not necessarily those of the NHS, the NIHR or the Department of Health and Social Care.

**Competing interests:** The authors have declared that no competing interests exist.

(clinical reasoning, dosage variables, exercise variables, evaluation criteria and progression) and 2 subthemes (progression criteria and progression variables). Excellent agreement and qualitative data supports exercise prescription complexity and the need for individualised, acceptable, and feasible exercise. Only 37% of important exercise components were generated from clinical trial data. Agreement was highest (88%-96%) for 3 dosage variables: intensity of effort, frequency, and repetitions.

## Conclusion

Multiple exercise and dosage variables are important, resulting in complex and individualised exercise prescription not found in clinical trials. Future research should use these important variables to prescribe an evidence-informed approach to exercise.

## Introduction

### Background

Neck pain is highly prevalent, accounting for 22% of all musculoskeletal disorders and is the 3rd leading cause of years lived with disability in the UK [1, 2]. Although exercise is the mainstay treatment for neck pain, there is still considerable uncertainty over the optimal content or delivery of exercise to maximise the effects of pain or disability [3–5]. In medicine, a dose-response relationship exists [6, 7], and if considering exercise as medicine, effects on pain and disability can be optimised by manipulating exercise dosage (sets, reps, loads, frequency etc.). Precision prescription of dosage and other variables such as exercise type, speed or order, increases the effectiveness of achieving physiological outcomes of hypertrophy, strength, power, endurance in healthy populations [8–17]. The effect of manipulating dosage and other exercise variables in neck pain populations on patient-reported outcome measures is less clear [18]. Experts agree that optimising neck pain exercise through further understanding of exercise and dosage variables is the leading research priority [4, 19].

Previous evidence synthesis and results from individual studies demonstrate a positive correlation between the exercise and dosage variables of duration, sets/repetitions, adherence and an improvement in neck pain [18, 20, 21]. Care must be taken however in transferring these findings to all neck pain exercise as different exercise interventions known to have different effects on spinal function were investigated (gymnastics, qigong, flexibility exercise and upper limb strengthening exercise) [22, 23].

A recent systematic review aimed to evaluate effectiveness and optimal dosage of different chronic non-specific neck pain (CNSNP) exercise programmes categorised by their intended effect on spinal function [24]. Exercises were categorised as motor control (submaximal effort exercises for the deep cervical muscles, improving co-ordination and sequential spinal control); segmental (exercises for the superficial cervical muscles improving the ability of the neck to produce, transfer and absorb force); pillar (exercises intended to develop the ability of the spine to maintain a neutral position) or upper limb (exercises intended to change the neuromuscular performance of the shoulder or shoulder girdle musculature) (Fig 1) [25]. Although the mechanisms by which exercise interventions improve pain or disability are unclear (e.g. exercise induced hypoalgesia, secondary to improvements in spinal function, secondary to improvements in physiological outcomes) systematic review findings demonstrate that the largest reduction in pain (moderate to very large short-term effects) were found for exercise training programmes combining both motor control and segmental exercises [24]. However,

**Motor Control**

**Definition:** Exercises intended to retrain co-ordination of cervical musculature or sequential segmental control of spinal movement using submaximal effort

**e.g.** Craniocervical flexion in supine; Craniocervical rotation in 4-point kneeling

**Segmental**

**Definition:** Exercises intended to develop the ability of the spine to endure the production, transference, or absorption of forces through the performance of sequential segmental movements

**e.g.** Cervical flexion using pulley system; Cervical flexion in supine; Cervical extension in 4-point kneeling

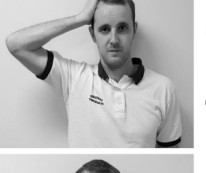
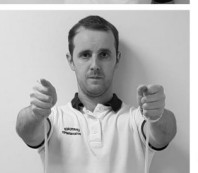

**Pillar**

**Definition:** Exercises intended to develop the ability of the spine to maintain a neutral position

**e.g.** Cervical isometric flexion/extension/lateral flexion using hand as resistance/resistance bands; Cervical isometric flexion against gravity in sitting

**Upper Limb**

**Definition:** Exercises intended to change the neuromuscular performance of the shoulder or shoulder girdle musculature

**e.g.** Resisted row; Shoulder press; Shrugs; Scapular retraction; Horizontal pull apart; Glenohumeral abduction with dumbbells

**Fig 1. Exercise classification with example exercises.**

evaluation of optimal motor control and segmental exercise and dosage variables was limited by inconsistent and poor intervention reporting; a finding not limited to CNSNP trials [26–30]. Although the Consensus on Exercise Reporting Template (CERT) and the Template for Intervention Description and Replication (TIDieR) have improved intervention reporting they currently do not provide guidance as to how or which exercise and dosage variables to report [31, 32].

Inadequate reporting of CNSNP exercise and dosage variables impacts evidence-informed practice, reducing treatment effectiveness causing ongoing disability [33]. Precision in reporting exercise interventions is needed to inform future research in CNSNP to investigate optimal exercise and dosage variables [24, 34]. Consensus on the important motor control and segmental exercise and dosage variables would strengthen the evidence, informing precision rehabilitation, therefore improving CNSNP.

## Aims of the study

1. To gain expert opinion on the important motor control and segmental exercise and dosage variables for CNSNP treatment.

2. To conduct a systematic process with experts to gain consensus on the important motor control and segmental exercise and dosage variables identified in Aim 1

## Methods

### Design

This international 3-round Delphi study was conducted between March—September 2020 according to an open-access protocol and is reported using Guidance on Conducting and REporting DElphi Studies (CREDES) [35, 36]. The University of Birmingham Ethics Committee granted ethical approval (REF: ERN_19–1857). The individual in Fig 1 has given written informed consent (as outlined in PLOS consent form) to publish their image. Written informed consent was received from all participants before completing any questionnaires. The consent process and each round was conducted electronically and anonymously using REDCap, a secure web application for building and managing online surveys [37, 38]. The study flow and objectives in each round are detailed in Fig 2 (see protocol for full methodological details rationale). Statements achieving consensus in round 3 identify the important exercise and dosage variables of an exercise training programme for CNSNP.

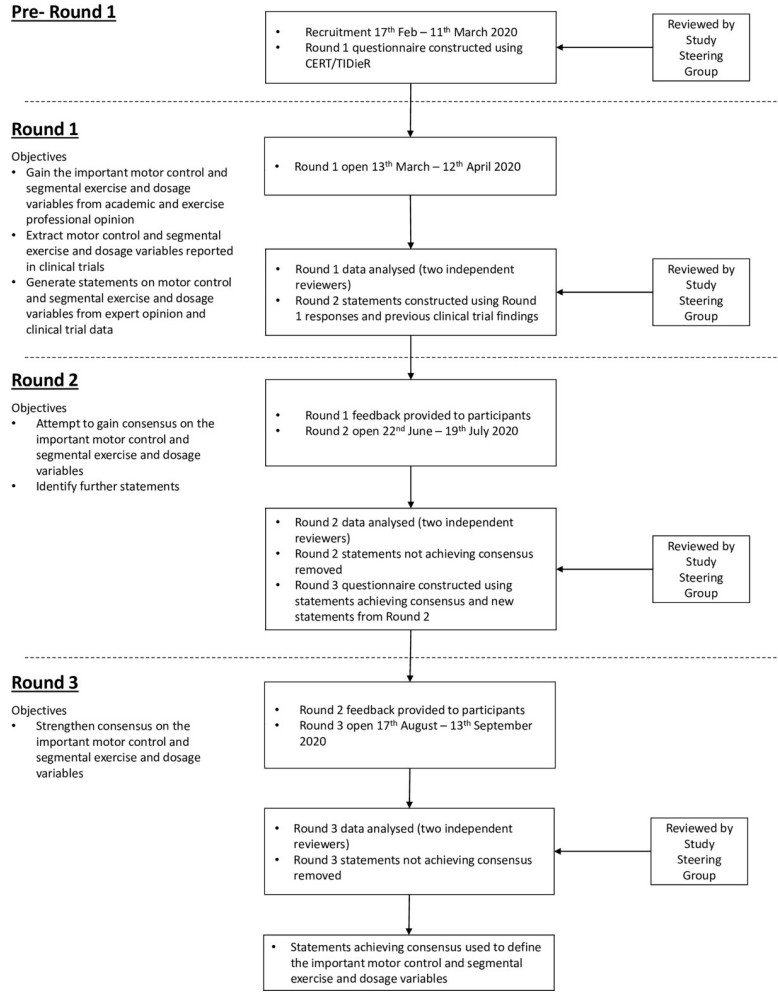

**Fig 2. Study flow and objectives.**

## Participants

Purposive sampling recruited experts in CNSNP exercise prescription from two populations, with no geographic limitations.

1. Academic experts having ≥2 peer-reviewed publications on CNSNP exercise in the past 10 years.

2. Clinical experts (physiotherapists, strength and conditioning coaches, osteopaths or chiropractors) with >5 years of experience or a postgraduate qualification in sports or musculoskeletal practice and treating ≥5 individuals with CNSNP per month using exercise.

## Recruitment

Recruitment from 17th February to 11th March 2020 identified experts through existing professional networks, Expertscape searches and CNSNP systematic reviews/randomised clinical trials indexed in PubMed, who were invited to participate by the lead author (J.P.) [19]. Experts were also recruited through social media calls and snowballing from other experts [39, 40]. Previous intervention development studies achieved consensus with 10–27 final round

responses [41–46] therefore we estimated a sample size of 40 experts was required to achieve a conservative estimate of 27 round 3 responses (70% response rate) [47]. Informed consent was obtained and the rights of participants were protected.

## Round 1

The round 1 questionnaire was developed following pilot testing on readability, relevance, and appropriateness through the Study Steering Group (SSG). Participant demographics (professional background, highest qualification, primary country of work, work setting) and academic/clinical expertise measures (H-index/peer-reviewed publication count and years' experience respectively) were collected [48, 49]. Before completing the questionnaire, the study team provided participants with a standardised summary of the systematic review results informing the Delphi study and with definitions and examples of motor control and segmental exercise [24]. They were then asked to list and explain important exercise and dosage variables for both motor control and segmental exercises when treating CNSNP and to provide factors informing their reasoning. Round 1 was open for 4 weeks.

During round 1 the study team (J.P./V.T.) also extracted exercise and dosage data from clinical trials describing motor control and segmental exercise cited in a recent systematic review [24]. Data was extracted independently and mapped to TIDieR/CERT [31, 32]. Previous trial data were collected concurrently during round 1 and combined with the data collected from experts before being analysed and presented back to the experts in round 2. Collecting clinical trial data and expert opinion data separately in round 1 improved content validity for future rounds [47, 50, 51]. Participant responses and clinical trial data were examined thematically by two independent researchers (J.P./V.T.) using QSR International's NVIVO 12 Plus software [52, 53]. Data was organised by participant as recommended when conducting thematic analysis on questionnaire responses [54]. Following a period of data familiarisation, data was coded iteratively. Codes were organised to develop candidate themes with central organising concepts both inductively, and then deductively informed by TIDieR/CERT. Candidate themes were edited to develop full themes by reviewing the codes and original questionnaire responses. Codes within themes were converted to statements. Any code described at least once was converted to a statement. Where multiple codes existed relating to a single statement, the code best-representing wording across participants was kept or a new encompassing statement was created [51]. Source data was stated at the end of each statement (i.e. expert opinion and/or clinical trial data). Complete agreement between researchers (J.P./V.T.) was required for themes and statements to be included [55].

## Round 2

Round 2 was developed from the themes and statements constructed from round 1. Study participants were provided with round 1 feedback on theme/statement generation and then asked to rate their agreement with statements in relation to motor control and segmental exercise separately using a 5-point Likert scale (1 = strongly disagree, 5 = strongly agree) [56]. Comments and further statements were provided in open text boxes. All participants were invited to participate in round 2 and demographic data was collected for those who had not completed round 1 [47]. Round 2 was open for 4 weeks.

Qualitative data were analysed thematically to construct new statements or edit/reword existing statements. Quantitative data analysis was completed by two researchers (J.P./V.T.) using IBM SPSS Statistics Version 25 using *a priori* criteria (see protocol for full details) [36, 57]. Expert consensus (the extent to which the group of experts share the same opinion) was evaluated for each statement using median ($\geq$3), interquartile range ($\leq$1.5) and percentage agreement (the percentage of responses rated agree/strongly agree)($\geq$60%) [42, 58]. For a

statement to be included in round 3, all three measures of expert consensus needed to be achieved. Statements failing to achieve consensus were removed for subsequent rounds. Consensus measures were used to evaluate each statement separately. To evaluate inter-expert agreement across multiple statements Kendall's Coefficient of Concordance was used with statistical significance set at P<0.05 [59]. Kendall's Coefficient of Concordance also produces a *W* value describing the strength of inter-expert agreement. Descriptive benchmarks have been recommended for measuring inter-rater agreement for ordinal data as <0.00 poor agreement, 0.00–0.20 slight agreement, 0.21–0.40 fair agreement, 0.41–0.60 moderate agreement, 0.61–0.80 substantial agreement, 0.81–1.00 almost perfect agreement [60].

### Round 3

Round 3 was constructed from new and existing statements achieving consensus in round 2. Previous round feedback was presented, and participants were asked to rate their agreement with statements as per round 2. Response clarification was invited using open text boxes, but new statements were not encouraged. All participants were invited, and demographic data were collected where missing. Round 3 was open for 4 weeks.

Quantitative data analysis followed the same procedure as round 2 using progressively increased criteria to encourage convergence and strengthen overall consensus (median ≥3.5, interquartile range ≤1, percentage agreement ≥70%) [42, 56]. Inter-expert agreement across all statements and within themes was evaluated using Kendall's Coefficient of Concordance (*W*)(P<0.05) [59]. Stability of the responses between rounds 2 and 3 was evaluated using the Wilcoxon rank-sum test (P<0.05) [58].

### Study steering group

The SSG consisted of the study researchers in addition to patients and external members with methodological and clinical expertise. The SSG met at key stages to provide study oversight on protocol design; participant recruitment; data analysis; study conduct and finding dissemination. Each round's questionnaire was piloted with the SSG testing readability, relevance, and appropriateness. Members of the SSG who were not co-authors did not have access to raw data or were able to influence the study process. Feedback and changes suggested by the SSG were agreed between the study co-authors before implementation.

### Patient and public involvement

Patient and public involvement (PPI) is defined by the National Institute for Health Research as research being carried out 'with' or 'by' patients or the public, rather than 'to', 'about' or 'for' them [61]. PPI was integral to this study through the SSG and their impact on the study is reported using the Guidance for Reporting Involvement of Patients and the Public 2 Short Form (GRIPP2-SF) [62].

## Results

Sixty-one experts were identified and contacted by the lead author (51 academics, 10 clinical experts) of which 18 expressed an interest to participate. A further 23 experts meeting eligibility requirements responded to snowballing/social media calls. Of the 41 experts interested, 37 provided consent and were enrolled onto the study.

### Participant demographics

Demographic data were collected for 34/37 participants representing 10 countries (Table 1). Demographic data were missing for 3 participants, 2 took part in round 2 and 1 failed to

**Table 1. Demographic details of participants.**

| Characteristics | Number of participants (n = 37) | Percentage of participants (%) |
|---|---|---|
| Professional background | | |
| Chiropractor | 1 | 2.70 |
| Osteopath | 0 | 0 |
| Physiotherapist | 32 | 86.49 |
| Strength and conditioning coach | 0 | 0 |
| Other: Human physiology and sports scientist | 1 | 2.70 |
| Not reported | 3 | 8.11 |
| Highest qualification | | |
| Doctor of Philosophy or equivalent | 15 | 40.54 |
| Master's degree | 8 | 21.62 |
| Bachelor's degree | 10 | 27.03 |
| Other: Post Graduate Diploma | 1 | 2.70 |
| Not reported | 3 | 8.11 |
| Country of work | | |
| Australia | 6 | 16.22 |
| Belgium | 2 | 5.41 |
| Canada | 2 | 5.41 |
| Denmark | 3 | 8.11 |
| France | 2 | 5.41 |
| Hong Kong | 1 | 2.70 |
| Ireland | 1 | 2.70 |
| South Africa | 1 | 2.70 |
| United Kingdom | 12 | 32.43 |
| United States | 2 | 5.41 |
| Not reported | 5 | 13.51 |
| Work setting* | | |
| Education | 17 | 45.95 |
| NHS/Public Health Service | 13 | 35.14 |
| Occupational health | 1 | 2.70 |
| Private | 14 | 37.84 |
| Research | 12 | 32.43 |
| Sports | 3 | 8.11 |
| Not reported | 3 | 8.11 |
| Expertise | | |
| Academic/education only | 3 | 8.11 |
| Clinical only | 15 | 40.54 |
| Both | 16 | 43.24 |
| Not reported | 3 | 8.11 |
| Academic expertise | | |
| H-Index | | |
| <10 | 2 | 10.53 [†] |
| 11–20 | 3 | 15.79 [†] |
| 21–30 | 3 | 15.79 [†] |
| 31–40 | 2 | 10.53 [†] |
| 40 + | 7 | 36.84 [†] |
| Rather not say | 2 | 10.53 [†] |
| Total peer-reviewed publications | | |

(*Continued*)

**Table 1.** (Continued)

| Characteristics | Number of participants (n = 37) | Percentage of participants (%) |
|---|---|---|
| ≤50 | 7 | 36.84 [†] |
| 51–100 | 4 | 21.05 [†] |
| 101–200 | 2 | 10.53 [†] |
| 200+ | 6 | 31.58 [†] |
| Rather not say | 0 | 0 [†] |
| Clinical expertise | | |
| Years qualified | | |
| ≤5 | 2 | 6.45 [‡] |
| 6–10 | 5 | 16.13 [‡] |
| 11–15 | 6 | 19.35 [‡] |
| 16–20 | 5 | 16.13 [‡] |
| 21+ | 13 | 41.94 [‡] |
| Work grade | | |
| Band 5/Junior | 0 | 0 [‡] |
| Band 6/Specialist | 4 | 12.90 [‡] |
| Band 7/Advanced | 8 | 25.81 [‡] |
| Band 8/Extended Scope or Consultant | 12 | 38.71 [‡] |
| Other | 7 | 22.58 [‡] |

[*] Participants may select more than 1 option.

[†] Percentages based on 19 participants identified as having academic expertise.

[‡] Percentages based on 31 participants identified as having clinical expertise.

complete any round. Participants were predominately physiotherapists (86%) and worked in a variety of clinical settings. Academic and clinical expertise was significant (37% of participants had an h-index >40; 32% had 200+ peer-reviewed publications, and 42% had been qualified for >21 years).

## Round 1

Twenty-nine out of 37 participants completed round 1 (response rate = 78%). A total of 442 codes were constructed from combining expert responses and clinical trial data. The complexity and diversity in responses warranted codes to be organised into 5 themes and 2 sub-themes outlining multiple components of exercise and dosage prescription (Table 2). Statements were developed for motor control exercise (n = 71), segmental exercise (n = 77) and the relationship between the two (n = 2). All motor control statements were also applicable to segmental exercise therefore 79 individual statements were generated (S1 Table).

## Round 2

Twenty-six out of 37 participants completed round 2 (response rate = 70%). Inter-expert agreement was statistically significant across all statements and across statements within themes (P<0.05) (Table 3). Consensus was achieved for 36/71 motor control statements, 50/77 segmental statements and 0/2 statements referring to the relationship between motor control and segmental exercise (S1 Table). The SSG agreed that participant comments supported 9 new statements (S1 Table) and rewording existing statements was not required. Participant comments suggested that the segmental exercise statement "*The amount of weight moved (Load) is an important variable to manipulate when making exercise harder*" should also be

**Table 2. Themes, central organising concepts, and supportive quotes.**

| Theme | Central organising concept | Supporting quotes |
|---|---|---|
| Theme 1: Clinical reasoning (22 statements) | There are various theories and principles that underpin the clinical reasoning when prescribing the parameters for exercise and dosage variables | *"These 2 categories of exercise are a continuum and while variables will differ depending on 'intended training effects i.e. skill acquisition v hypertrophy v endurance etc' it really does depend on the persons life requirements."* (Expert Opinion–R1ID5) |
| | | *"Giving a theoretical framework sticking to patient's values and expectations."* (Expert Opinion–R1ID1) |
| | | *"Depends on multiple factors: Pain levels, quality of movement, technique during the exercise, general activity levels (i.e. sedentary v's rugby prop forward) etc"* (Expert Opinion–R1ID25) |
| | | *"I will be guided by the patient as to what they believe is an achievable and realistic expectation of how much exercise can be performed whilst continuing with the other aspects of their life."* (Expert Opinion–R1ID8) |
| | | *"Generally the exercise is guided by several factors and should not be a one size fits all approach."* (Expert Opinion–R1ID7) |
| Theme 2: Dosage variables—how much? (12 Statements) | The variables that describe how much exercise will be completed | *"Number of sessions through the day (more is better, at least 3x is encouraged)."* (Expert Opinion–R1ID3) |
| | | *"This could be higher reps, less resistance e.g 3 sets of 10–15 reps or higher resistance and less reps 6–8, 2–3 sets."* (Expert Opinion–R1ID18) |
| | | *"...40%-60% of effort while performing exercises."* (Expert Opinion–R1ID9) |
| Theme 3: Exercise variables—how? (9 Statements) | The variables that describe how exercises should be performed | *"... Speed of exercise can be varied depending on the patient, as can length of hold, degree of movement (i.e. half range, 3/4 range) and position of exercise (i.e. lying, sitting, standing etc)..."* (Expert Opinion–R1ID25) |
| | | *"...then I'll tend to use an attentional focus on other sensations..."* (Expert Opinion–R1ID1) |
| Theme 4: Evaluation criteria (11 statements) | The factors that exercise professionals can use to evaluate whether exercise and dosage variables are appropriate and whether they should be adapted | *"...exercises...were performed without any provocation of neck pain and with performance of smooth uniplanar movements."* (Clinical Trial data) |
| | | *"Reps/sets can be reduced if fatigue is brought on sooner."* (Expert Opinion–R1ID24) |
| | | *"...is the patient in a job/lifestyle where they can do what I am asking —if not we might have to adapt what I am asking."* (Expert Opinion–R1ID3) |
| Theme 5: Progressive overload (2 statements) | Exercise and dosage variables are gradually changed to increase the difficulty of the exercise | *"...explain the overload principle to clients and that they need to do more than what they normally do."* (Expert Opinion–R1ID13) |
| Subtheme 5.1: Progression criteria—when to make exercise harder? (10 statements) | The criteria that can be used to identify when exercise should be made more difficult by adapting exercise and dosage variables | *"Progression to a higher pressure level occurs when the patient can successfully perform the 10x10 sec holds at the prescribed level."* (Expert Opinion–R1ID6) |
| | | *"...once this gets easier I will then begin to encourage them to do bigger sets but less frequently."* (Expert Opinion–R1ID3) |
| | | *"However I would not insist on strictly adhering to the given dosage only and allow patient to judge and progress accordingly"* (Expert Opinion–R1ID9) |
| Subtheme 5.2: Progression variables—How to make exercise harder? (13 statements) | The exercise and dosage variables that should be changed to make exercise more difficult | *"Performance sports may progress to resistance band or sand bag forehead weight."* (Expert Opinion–R1ID20) |
| | | *"...altering the position they perform it in to make it more difficult."* (Expert Opinion–R1ID7) |
| | | *"Tempo should still be considered but can be used to alter the level of difficulty, execution should still be good technique."* (Expert Opinion–R1ID7) |
| | | *"Repetitions: 8–12 Sets: 2–3 Intensity: 60%-70% of 1 RM ((commence with 40% of 1 RM and progress gradually))"* (Expert Opinion–R1ID9) |

**Table 3. Inter-expert agreement using Kendall's Coefficient of Concordance.**

| | Round 2 | | | | | | Round 3 | | | | | |
|---|---|---|---|---|---|---|---|---|---|---|---|---|
| | All Categories | | Motor Control | | Segmental | | All Categories | | Motor Control | | Segmental | |
| | *W* | p | *W* | p | *W* | p | *W* | p | *W* | p | *W* | p |
| **All Statements** | 0.335 | <0.001 | 0.268 | <0.001 | 0.361 | <0.001 | 0.260 | <0.001 | 0.245 | <0.001 | 0.286 | <0.001 |
| **Theme 1** | 0.435 | <0.001 | 0.441 | <0.001 | 0.469 | <0.001 | 0.269 | <0.001 | 0.258 | <0.001 | 0.293 | <0.001 |
| **Theme 2** | 0.216 | <0.001 | 0.239 | <0.001 | 0.209 | <0.001 | 0.191 | <0.001 | 0.162 | 0.002 | 0.224 | <0.001 |
| **Theme 3** | 0.412 | <0.001 | 0.245 | <0.001 | 0.503 | <0.001 | 0.189 | <0.001 | 0.004 | 0.763 | 0.286 | <0.001 |
| **Theme 4** | 0.222 | <0.001 | 0.220 | <0.001 | 0.228 | <0.001 | 0.215 | <0.001 | 0.204 | <0.001 | 0.227 | <0.001 |
| **Theme 5** | 0.304 | <0.001 | 0.388 | 0.002 | 0.333 | 0.004 | NA | NA | NA | NA | NA | NA |
| **Theme 5 Subtheme 1** | 0.182 | <0.001 | 0.199 | <0.001 | 0.184 | <0.001 | 0.265 | <0.001 | 0.241 | <0.001 | 0.311 | <0.001 |
| **Theme 5 Subtheme 2** | 0.129 | <0.001 | 0.078 | 0.034 | 0.143 | <0.001 | 0.213 | <0.001 | 0.142 | 0.011 | 0.271 | <0.001 |

Kendall's Coefficient of Concordance measures inter-expert agreement across multiple statements. Statistically significant inter-expert agreement was measured using $p < 0.05$. Strength of inter-expert agreement *(W) measured* using <0.00 poor agreement, 0.00–0.20 slight agreement, 0.21–0.40 fair agreement, 0.41–0.60 moderate agreement, 0.61–0.80 substantial agreement, 0.81–1.00 almost perfect agreement

applicable for motor control exercise. A total of 61 statements (46 motor control; 59 segmental) were taken into Round 3.

## Round 3

Twenty-four out of 37 participants completed round 3 (response rate = 65%). Inter-expert agreement was statistically significant across all statements and across statements within themes (P<0.05) except Theme 3 motor control statements ($W = 0.004$; P = 0.763) (Table 3). Consensus statistics were stable between rounds 2 and 3 (Wilcoxon rank-sum test P>0.05). A total of 46 statements achieved consensus in round 3 (33/46 motor control; 45/59 segmental) of which 63% were generated from expert opinion and not clinical trial data (Table 4). Of the statements achieving consensus, statement percentage agreement (percentage of experts rating agree/strongly agree) was high ranging from 70% to 100%. The important components of prescribing motor control and segmental exercise and dosage variables for CNSNP treatment are outlined in Table 4.

## Patient and public involvement

PPI as part of the SSG is reported in (Table 5) [62].

## Discussion

This is the first study to gain international expert consensus on exercise and dosage variables for CNSNP. Expert consensus supports 33 motor control and 45 segmental exercise statements, with 5 themes outlining the multiple important interacting components of exercise and dosage prescription. Researchers and clinicians should consider using these themes and statements to guide the prescription of CNSNP exercise in future research and clinical practice.

## Individualisation

For exercise prescription to be precise for all patients, it needs to be individualised, the importance of which was highlighted within this study. Excellent percentage agreement for the statement *"it is important that exercise and dosage variables are tailored to each patient"* supports this concept (motor control 95.83%; segmental 100%). Also, all statements achieving

**Table 4. Statements achieving consensus post round 3.**

| | Statement Description | Motor Control | | | Segmental | | |
|---|---|---|---|---|---|---|---|
| | | Median | IQR | % | Median | IQR | % |
| **Theme 1 –Clinical reasoning** | | | | | | | |
| 1 | Variables are tailored to each patient* | 5.00 | 1.00 | 95.83 | 5.00 | 0.75 | 100.00 |
| 2 | Variables are kept simple* | 5.00 | 1.00 | 95.83 | 5.00 | 0.75 | 100.00 |
| 3 | Patient considers variables to be realistic* | 5.00 | 0.75 | 95.83 | 5.00 | 0.00 | 100.00 |
| 4 | Variables are an achievable challenge* | 5.00 | 1.00 | 95.83 | 5.00 | 1.00 | 100.00 |
| 5 | Variables are acceptable to the patient* | 5.00 | 1.00 | 87.50 | 5.00 | 1.00 | 91.67 |
| 6 | Exercise is not time consuming to complete* | 4.00 | 1.00 | 79.17 | 4.00 | 1.00 | 87.50 |
| 7 | Variables are expected to be adhered to* | 4.00 | 0.00 | 79.17 | 4.00 | 0.75 | 83.33 |
| 8 | Variables are prescribed in collaboration with the patient* | 4.50 | 1.00 | 87.50 | 5.00 | 1.00 | 95.83 |
| 9 | Variables are prescribed specifically to change neuromuscular function or motor capacity based on assessment findings and patients' functional goals or demands* | 4.00 | 1.00 | 79.17 | 4.00 | 1.00 | 79.17 |
| 10 | Variables are prescribed collaboratively with patients within a framework that is sufficient to affect neuromuscular performance* | 5.00 | 1.00 | 91.67 | 5.00 | 1.00 | 91.67 |
| 11 | Patients are educated so they understand and accept the rationale for the variables prescribed* | 5.00 | 0.75 | 91.67 | 5.00 | 1.00 | 95.83 |
| 12 | Variables are monitored by patients and adapted independently* | 4.00 | 0.75 | 75.00 | 4.00 | 0.75 | 79.17 |
| 13 | Variables are adapted for different stages of rehabilitation process* | 5.00 | 0.00 | 95.83 | 5.00 | 1.00 | 100.00 |
| 14 | Variables are prescribed to specifically improve patient reported symptoms* | 4.00 | 0.75 | 75.00 | - | - | - |
| 15 | Variables are prescribed based on equipment availability* | - | - | - | 4.50 | 1.00 | 79.17 |
| **Theme 2 –Dosage variables–how much?** | | | | | | | |
| 1 | Frequency† | 4.00 | 1.00 | 87.50 | 4.00 | 1.00 | 95.65 |
| 2 | Intensity of effort† | 4.00 | 1.00 | 83.33 | 4.00 | 1.00 | 95.83 |
| 3 | Repetitions† | 4.00 | 0.75 | 79.17 | 4.00 | 1.00 | 91.30 |
| 4 | Sets† | 4.00 | 0.00 | 79.17 | 4.00 | 0.75 | 87.50 |
| 5 | Load† | - | - | - | 4.00 | 1.00 | 87.50 |
| 6 | Duration of exercise training programme† | - | - | - | 4.00 | 0.75 | 79.17 |
| **Theme 3 –Exercise variables–how?** | | | | | | | |
| 1 | Exercise position* | 4.00 | 1.00 | 87.50 | 4.00 | 1.00 | 91.67 |
| 2 | Range of movement† | - | - | - | 4.50 | 1.00 | 83.33 |
| 3 | Direction of resistance* | - | - | - | 4.00 | 1.00 | 87.50 |
| **Theme 4 –Evaluation criteria** | | | | | | | |
| 1 | Technique during exercise† | 5.00 | 1.00 | 91.67 | 4.00 | 1.00 | 95.83 |
| 2 | Patient effort† | 4.00 | 1.00 | 91.67 | 5.00 | 1.00 | 95.83 |
| 3 | Patient compliance* | 5.00 | 1.00 | 91.67 | 5.00 | 1.00 | 95.83 |
| 4 | Pain during exercise† | 4.00 | 1.00 | 83.33 | 4.00 | 1.00 | 83.33 |
| 5 | Pain after exercise* | 4.50 | 1.00 | 87.50 | 4.50 | 1.00 | 91.67 |
| 6 | Fatigue during exercise* | 4.00 | 1.00 | 87.50 | 4.00 | 1.00 | 91.67 |
| 7 | Fatigue after exercise* | 4.00 | 1.00 | 70.83 | 4.00 | 0.00 | 79.17 |
| **Theme 5 –Progressive overload** | | | | | | | |
| 1 | Variables are progressively increased over time to make exercise harder† | - | - | - | 5.00 | 1.00 | 100.00 |
| **Theme 5.1 –Progression criteria—when to make exercise harder?** | | | | | | | |
| 1 | When a patient no longer perceives exercise to be difficult* | 4.00 | 1.00 | 91.67 | 4.50 | 1.00 | 95.83 |
| 2 | When functional goals improve* | 4.00 | 1.00 | 78.26 | 4.00 | 1.00 | 86.96 |
| 3 | When neuromuscular performance has improved based on objective findings* | 4.00 | 1.00 | 83.33 | 4.00 | 1.00 | 79.17 |
| 4 | When a patient feels they are ready to do so* | - | - | - | 4.00 | 1.00 | 87.50 |
| 5 | When patients no longer fatigue during exercise* | - | - | - | 4.00 | 1.00 | 82.61 |
| 6 | When patient symptoms decrease* | - | - | - | 4.00 | 1.00 | 70.83 |
| **Theme 5.2—Progression variables—How to make exercise harder?** | | | | | | | |

*(Continued)*

**Table 4.** (Continued)

| | Statement Description | Motor Control | | | Segmental | | |
|---|---|---|---|---|---|---|---|
| | | Median | IQR | % | Median | IQR | % |
| 1 | The variable most pertinent to a patient's functional activity* | 5.00 | 1.00 | 91.67 | 5.00 | 1.00 | 95.65 |
| 2 | Range of movement[†] | 4.00 | 0.75 | 75.00 | 4.00 | 1.00 | 91.67 |
| 3 | Repetitions[†] | 4.00 | 1.00 | 79.17 | 4.00 | 1.00 | 83.33 |
| 4 | Sets[†] | 4.00 | 0.00 | 83.33 | 4.00 | 0.75 | 79.17 |
| 5 | Load[†] | - | - | - | 5.00 | 1.00 | 100.00 |
| 6 | Intensity of effort[†] | - | - | - | 4.50 | 1.00 | 91.67 |
| 7 | Frequency[†] | - | - | - | 4.00 | 1.00 | 86.96 |
| 8 | Exercise position* | - | - | - | 4.00 | 0.00 | 83.33 |

IQR, Interquartile range; %, Percentage agreement–The percentage of experts rating the statement as agree or strongly agree.

* Indicates a statement constructed from expert opinion data only.

[†] Indicates a statement constructed from expert opinion & clinical trial data.

consensus in Themes 1, 4 and 5.1 result in individualised exercise prescription. Participants open text responses identified that the importance of exercise and dosage variables may differ depending on patient characteristics. This variability between patients is further supported by the slight to fair inter-expert agreement ($W$ = 0.142 to 0.311), despite achieving statistical significance ($p < 0.05$).

Although the heterogenic nature of neck pain warrants tailored interventions, this aspect of clinical reasoning is lacking in neck pain trials [63]; and is largely limited to individualising exercise intensity or range of movement [24, 64]. While other spinal research demonstrates the benefits of individualised treatment [65], Svedmark et al., 2016 found no difference in pain and disability between an individualised exercise programme tailored to neuromuscular dysfunction versus a non-individualised exercise programme [66]. Although this could suggest individualised exercise may not be beneficial the study did not evaluate the effect of the exercise interventions on the neuromuscular dysfunction on which it was based. Therefore, it is unknown whether there was no difference in pain or disability because of no improvement in the neuromuscular dysfunction the exercise programme intended to improve or because this degree of individualisation is not beneficial. While the true effectiveness of individualising neck pain exercise is unclear, patients with spinal pain report this is an important component of their care [67]. Our results support the importance of individualising motor control and segmental exercise and dosage variables but further research evaluating effectiveness is required.

### Acceptability and feasibility

Excellent statement percentage agreement (segmental = 100%; motor control = 95.83%) supports exercise and dosage variables being simple, realistic, prescribed collaboratively with patients and an achievable challenge (Theme 1, Statements 2,3,4,8). Statement percentage agreement was also high for acceptability and adherence statements (Theme 1, Statements 5,6). Collectively these statements outline exercise acceptability and feasibility of which there is a paucity of research within the CNSNP exercise literature. Further understanding of patients with CNSNPs' experiences of exercise will inform prescription so that it is acceptable and feasible; improving adherence and benefitting outcomes [68, 69]. These results demonstrate the importance of prescribing acceptable and feasible exercise, but further research evaluating these concepts is warranted.

**Table 5. GRIPP2 SF reporting patient and public involvement.**

| 1 Aim |
| --- |
| To collaboratively involve patients with experience of performing exercise interventions for neck pain in the design, conduct and dissemination of the Delphi study |

| 2 Methods |
| --- |
| Two patients were recruited to the SSG to provide oversight at all stages of study development, conduct and reporting. Patient representatives were involved in the following stages: |

1. Systematic review findings were summarised and presented to the patients (and SSG) using lay language by the study team for comments
2. One patient representative acted as Co-chair of the SSG facilitating meeting agendas and encouraging other patients to be actively involved in discussions
3. Developing the project aims and objectives
4. Developing expert eligibility criteria
5. Piloting and feedback on invitation emails and questionnaires at each round
6. Reviewed of themes and statements providing alternative wording where statements were not clear
7. Reviewed summary of findings
8. Dissemination plans
9. Advice and suggestions regarding future research implications

Patient representatives were not involved in the consensus procedures.

| 3 Study results |
| --- |
| Patient representatives had a positive impact on this study in multiple ways. |

1. Original drafts of the systematic review informing this study used the term "Resistance Training" to collectively describe the exercises included in the Delphi study. When systematic review findings were presented to the patient representatives (Method, Point 1) they recommended the term "exercise training programme" rather than resistance training. Patients felt resistance training had a strong association with exercises performed in a gym or fitness centre and believed exercise training programme was more appropriate for exercise that could be conducted in any location (i.e. home, gym, clinic). As a result of our PPI, the term exercise training programme has been used in this manuscript.

2. When reviewing a draft of the Delphi study protocol, patient representatives recommended international experts with clinical expertise should also be recruited to better reflect an international consensus process. Therefore international experts with clinical expertise were included in this study.

3. Patient representatives helped design the questionnaires presented to experts at each round. Patients made suggestions to reduce content and improve the clarity of the information presented to the experts (e.g., the use of page numbers, suggesting estimated completion times for each round and themes)

4. Upon completion of the Delphi study, the results were presented to the patient representatives. Patients were pleased that statements regarding exercise acceptability and feasibility achieved consensus and felt strongly that future research should evaluate these concepts. The patients also suggested that acceptability and feasibility may be different between individuals and future research should take this into consideration when evaluating acceptability/feasibility. As a result of our PPI, acceptability and feasibility of exercise is explored in the discussion section of this manuscript. We plan to conduct further research to evaluate acceptability/feasibility of exercise.

5. Upon completion of the Delphi study, the results were presented to the patient representatives. Patient representatives highlighted their importance of being able to self-regulate exercise. Although this concept achieved consensus, the statement agreement was lower than other statements, suggesting expert and patient opinions on important components of exercise prescription may differ. Patients encouraged future research to evaluate patients' needs and desires when prescribing exercise as these maybe different to clinicians/experts. As a result of our PPI, we intend to conduct research to investigate the needs of patients that exercise must satisfy, and to develop an exercise programme where patients can self-regulate and monitor exercise and dosage variables.

6. Upon completion of the Delphi study, patient representatives also provided recommendations as to how the findings of this study, and future research should be disseminated to the public and patients. Patient representatives felt it was important that during treatment clinicians should make patients aware that healthcare professionals may not "*have all the answers*" regarding exercise and there are gaps in the literature which require research. They felt this information would 1) help manage patient's expectations 2) appreciate exercise requires tailoring and modifications before "*getting it right*" and 3) encourage other patients to take part in future research or PPI activities. As a result of our PPI, our dissemination activities will include 1) educating clinicians to discuss the degree of uncertainty regarding optimal exercise prescription when treating patients, to help manage expectations and encourage collaborative exercise prescription 2) updating our PPI literature to make it clear that there are uncertainties regarding exercise prescription requiring research that patients can be involved in.

No negative outcomes were identified from involving patients in this study.

| 4 Discussion and conclusions |
| --- |
| Patient involvement influenced this study and future projects in important ways. The degree to which patients impacted on the study is largely due to their unique perspective on exercise prescription, having both managed neck pain symptoms with exercise. Additionally, both patient representatives have previously been involved in research/academia (1 is study for a PhD in agricultural law, 1 works as a personal assistant in an academic institution) and understood the importance of their perspectives in designing and conducting research. |

Furthermore, having a patient as co-chair for the study steering group, resulted in increased engagement during meetings. Unfortunately, as this study was conducted during the COVID-19 pandemic, meetings in the later stages of the study were prevented by redeployment of the lead author. As a result, communication between the lead author and patients was largely conducted by email and patient representatives felt they were unable to contribute as significantly as face-to-face meetings. Considering this, the final steering group was conducted using 1-2-1 video conferencing to enable greater interaction and contribution. While this resulted in better engagement, it was not as good as the discussions that occurred when patients were able to interact with each other as in the early study steering group meetings.

| 5 Reflections/critical perspective |
| --- |
| Key strengths of the patient involvement in this study was 1) involvement from an early stage through to dissemination, 2) acting as co-chair and 3) having direct experience of completing exercise for neck pain. Future studies should look to engage a diverse representation of patients in a group setting as this resulted in the most engaging and productive discussions. |

## Exercise interventions in clinical trials are substandard

Sixty-three percent of the statements achieving consensus in round 3 were generated from expert opinion alone and not from clinical trial data, including those around individualisation, acceptability, and feasibility. Findings demonstrate that motor control and segmental exercise in clinical trials is not prescribed using the components experts agree are important. Although the protocol-driven approach to exercise prescription in clinical trials may improve treatment fidelity and consistency between patients, it fails to acknowledge the complex requirements of exercise, potentially explaining the modest effect sizes seen [3–5]. Consensus findings also demonstrate the complexity of exercise for CNSNP, supported by the large volume of statements achieving consensus and the interacting components within and between themes, in particular Themes 1, 4 and 5. The effectiveness of exercise interventions embracing complexity, utilising the important components of motor control and segmental exercise prescription agreed through expert consensus has not been evaluated in a clinical trial [24]. To satisfactorily test this type of complex exercise, adequate preliminary intervention development is required. To achieve this future research must develop and optimise motor control and segmental exercise for CNSNP using the MRC complex intervention development guidance prior to phase III testing [70].

## Exercise type differences

Results support different approaches to exercise prescription for motor control and segmental exercise. Over 30% of statements were only applicable to either motor control or segmental exercise, and statement percentage agreement was higher for segmental exercise in 27/32 statements applicable to both exercise types. Exercise type disparity is consistent with literature demonstrating differing physiological effects, suggesting a one-size-fits-all approach to exercise prescription is not appropriate [22, 23]. This also questions the validity of combining exercise types during evidence synthesis and justifies the need for precision prescription [18].

## Exercise and dosage variables

Our results found consensus for the variables *"frequency"*, *"repetitions"*, *"sets"*, *"intensity of effort"* and *"exercise position"* for both exercise types and *"load"*, *"duration of training programme"*, *"range of movement"* and *"direction of resistance"* for segmental exercise. Statement percentage agreement was highest for frequency, intensity of effort and exercise position, and was higher for segmental exercise than motor control exercise. Little research has evaluated the optimal values for these variables when prescribing motor control or segmental exercise for CNSNP.

Motor control and segmental exercise frequency varies in trials (x3 daily to x3 weekly). Although our evidence synthesis suggests increased frequency of motor control exercise may increase effectiveness, the true effect of motor control or segmental exercise frequency is unknown as the observed effects may be attributed to increase in volume (how much exercise is done), rather than frequency [18, 24]. To the authors' knowledge the desired intensity of effort for motor control exercise has not been reported although segmental exercise has been prescribed using 6/10 Borg [71]. Although the Borg CR-10 scale is a reliable and valid method to evaluate and monitor training load, there has been limited research evaluating its use for neck exercise [72, 73]. Care must be taken using findings from a healthy population as evidence from patients with neck pain report higher intensities of effort compared to a healthy population using the same load [74]. Although our results support the importance of various exercise and dosage variables, the optimal values to achieve the desired outcomes (physiological or patient-reported) in a neck pain population are unknown and merit further

investigation. Due to the need for individualised care, it is likely that optimal values to achieve desired outcomes will vary between individuals but will also vary between different time points for the same individual.

**Strengths and limitations.** This study was conducted according to a published protocol, stating *a priori* criteria for consensus, and is reported using CREDES [35, 36]. Statements were constructed from expert opinion and clinical trial data, reflecting both clinical practice and research. Using open questions in round 1 improved content validity as statements were generated by expert opinion [47, 50]. Introducing statements from clinical trials in round 2 reduced experimenter bias [51]. Use of the SSG that included patients, clinical and methodological expertise providing independent oversight of study conduct and analysis.

Although a diverse range of countries was represented, a large proportion of participants were based within the UK limiting generalisability. We were unable to recruit strength and conditioning experts. Strength and conditioning coaches responding to invites failed to meet eligibility as they saw very few patients with CNSNP, questioning their expertise in exercise prescription for CNSNP specifically. Despite the sports setting being underrepresented (8.11%), there was significant expertise in CNSNP exercise prescription amongst the included participants. The round 3 response rate was lower than anticipated, however, previous interventions have been developed using much smaller sample sizes [43–46].

## Research and clinical implications

Results emphasise the importance of individualising exercise prescription that is acceptable and feasible to patients. Exercise professionals should ensure when prescribing motor control and segmental exercise that frequency, intensity of effort and exercise position are outlined. However, the optimal parameters for these variables are unknown and warrant further investigation.

Future clinical trials should evaluate exercise interventions that reflect the important components of exercise prescription outlined. Due to its complexity, exercise interventions for CNSNP should be developed using the MRC complex intervention development framework [70]. The PPI involved in this study highlighted the importance of being able to self-regulate exercise and dosage variables to patients. Although this concept achieved consensus amongst experts, percentage agreement was lower than other statements, suggesting experts and patients have different opinions on important components of exercise prescription. Future research must use range of data sources, including patient opinion, to develop an evidence-informed approach to exercise prescription for future trials.

## Conclusion

This Delphi outlines the multiple important interacting components of exercise and dosage prescription, of which very few were reported within clinical trials. Although consensus was achieved for the importance of variables such as frequency, intensity of effort and exercise position it is important that exercise and dosage prescription in future research is individualised, acceptable and feasible. Future work should acknowledge the complexity of exercise as a management intervention and develop an evidence-informed framework to guide exercise and dosage prescription for research and clinical practice.

## Supporting information

**S1 Table. Statement consensus rounds 2 and 3.** Abbreviations: IQR: Interquartile range; %: Percentage agreement–The percentage of experts rating the statement as agree or strongly

agree; ✗: round consensus criteria not achieved;✓ round consensus criteria achieved.
(PDF)

**S1 Dataset. Anonymised raw data for all rounds.**
(XLSX)

## Acknowledgments

The authors would like to thank all members of the Study Steering Group for study oversight. The authors would also like to thank all expert participants who took part in the study. Finally, the lead author would like to thank the redeployed therapy team for their assistance in caseload management to allow continued work on this study.

## Author Contributions

**Conceptualization:** Jonathan Price, Alison Rushton, Vasileios Tyros, Nicola R. Heneghan.

**Data curation:** Jonathan Price.

**Formal analysis:** Jonathan Price, Vasileios Tyros.

**Investigation:** Jonathan Price, Vasileios Tyros.

**Methodology:** Jonathan Price, Alison Rushton, Vasileios Tyros, Nicola R. Heneghan.

**Project administration:** Jonathan Price.

**Resources:** Jonathan Price, Alison Rushton, Nicola R. Heneghan.

**Supervision:** Alison Rushton, Nicola R. Heneghan.

**Validation:** Jonathan Price, Alison Rushton, Nicola R. Heneghan.

**Visualization:** Jonathan Price.

**Writing – original draft:** Jonathan Price.

**Writing – review & editing:** Jonathan Price, Alison Rushton, Vasileios Tyros, Nicola R. Heneghan.

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
