## [Decision Letter · Decision Letter 0]

21 Apr 2021

PONE-D-21-00717

Expert consensus on the important chronic non-specific neck pain motor control and segmental exercise and dosage variables: an international e-Delphi study

PLOS ONE

Dear Dr. Price,

Thank you for submitting your manuscript to PLOS ONE. After careful consideration, we feel that it has merit but does not fully meet PLOS ONE’s publication criteria as it currently stands. Therefore, we invite you to submit a revised version of the manuscript that addresses the points raised during the review process.

Please address the serious questions raised by the reviewers. 

We look forward to receiving your revised manuscript.

Kind regards,

Zubing Mei, MD,PH.D

Academic Editor

PLOS ONE

Journal Requirements:

Please provide additional details regarding participant consent. In the ethics statement in the Methods and online submission information, please ensure that you have specified what type you obtained (for instance, written or verbal, and if verbal, how it was documented and witnessed). If your study included minors, state whether you obtained consent from parents or guardians. If the need for consent was waived by the ethics committee, please include this information.

Please correct your reference to "p=0.000" to "p<0.001" or as similarly appropriate, as p values cannot equal zero

We note that Figure 1 includes an image of a  participant in the study.

Reviewers' comments:

Reviewer's Responses to Questions

**Comments to the Author**

1. Is the manuscript technically sound, and do the data support the conclusions?

Reviewer #1: Yes

Reviewer #2: Yes

Reviewer #3: Yes

2. Has the statistical analysis been performed appropriately and rigorously? 

Reviewer #1: Yes

Reviewer #2: Yes

Reviewer #3: Yes

3. Have the authors made all data underlying the findings in their manuscript fully available?

Reviewer #1: No

Reviewer #2: Yes

Reviewer #3: Yes

4. Is the manuscript presented in an intelligible fashion and written in standard English?

Reviewer #1: Yes

Reviewer #2: Yes

Reviewer #3: Yes

5. Review Comments to the Author

Reviewer #1: More attention to make clinical practice a better part of the methods of scientific research is extremely important to me in order to better translate the results found into our practice (EBP). Your research makes an important contribution to this.

I am happy to read that experts believe that exercises should match with the patient. Unfortunately, this does not appear to be the case with interventions in general in patients with non-specific neck pain. RCTs investigate specific interventions such as exercises in a heterogeneous mismatched population as my own research showed. (The clinical reasoning process in randomized clinical trials with patients with non-specific neck pain is incomplete: A systematic review.)

Because clinical reasoning in scientific research is close to my heart, I have given my additional opinion here and there. I don't expect you to do anything with that.

Introduction

What remains unclear is what the exercises are for. On line 39 you talk about physiological constructs such as strength and endurance. Similarly on line 52 the construct spinal function. However, on lines 41 and 42 you speak of neck pain exercises with the underlying construct exercise-induced hypoalgesia (ref18).

So what I wonder is whether the purpose of the exercise is to improve more physiological constructs as you explain from line 52 or whether the exercise should not improve a spinal function as long as it leads to exercise-induced hypoalgesia.

This distinction seems important to me since for the first group the patient must have a function limitation, such as for example a reduced motor control or endurance or strength, while the second group of “Hypoalgesia” only needs to have pain. This distinction may in turn influence dosage variables since it seems important that the intervention is appropriate for the right group of patients. (Therapeutic validity and effectiveness of preoperative exercise on functional recovery after joint replacement: a systematic review and meta-analysis. Hoogeboom TJ, Oosting E, Vriezekolk JE, Veenhof C, Siemonsma PC, de Bie RA, van den Ende CH, van Meeteren NL.PLoS One. 2012;7(5):e38031. doi: 10.1371/journal.pone.0038031)

Therefore, with regard to the goals of the study it remains unclear to me whether the purpose of the exercises were improving co-ordination (motor control) and improving the ability of the neck to produce, transfer and absorb force (segmental exercise) (lines 52-55) after which, if this improved, pain reduction may occur or was the purpose of the exercises pain reduction regardless of whether a function improved or both.

Please clarify the primary purpose of the exercises.

Method

The method is clearly described. The use of text boxes gives this study qualitatively more depth than a Delphi study in which one only scores on consensus.

It is a good thing that patients (page 18 item 4) also indicate that the intervention should be tailored to each person. I translate that if the exercise should aim to improve the present disability. Here, too, the purpose of the exercise appears to be important.

Results

Since I am not from the UK I do not understand what is meant by band 5 etcetera at work grade.

Line 209 Table 3 and line 222 again a table 3. Would you please correct this?

Line 221 A result of round 3 is mentioned in round 2.

Discussion

Line 246 I would be reluctant to consider the variables about which there is consensus as the truth. Further research should show whether the variables found really matter, I think. Although I think you are right ...

(A Delphi Study: Defining a Unicorn. Weisman A, Meakins A, Rotem-Lehrer N.Pain Med. 2018 Jun 1;19(6):1295. doi: 10.1093/pm/pnx327.)

Line 260 You stated: “However, the lack of effect maybe due to inadequacies in the clinical reasoning model used to identify appropriate treatment”. This may be, however, they assess the effect with PROMs without first determining whether the intervention has improved the impairment. For example their impairment "reduced cervical flexibility" remains unclear whether the cervical flexibility has been improved. Improving flexibility alone does not necessarily lead to an experienced improvement measured with an NPRS or NDI. They did not use the correct outcome measure measuring the primary goal of the intervention.

Ah on line 325 you will discuss the use of outcome measures. I think we give exercises to improve physiological variables. It is therefore important to first determine whether a physiological variable has improved before it can be expected that the patient will experience improved thanks to the improved physiological variable ... In this way we can gain insight into whether the patient has improved thanks to or despite the exercises.

That is why, like you, I still believe that interventions should be individualized.

Line 283 I agree with your statement on protocol driven. In addition, exercises are investigated in RCTs in heterogeneous populations without it being clear whether the patients have the limitation that matches the exercise.

Kendall's Coefficient of Concordance measures inter-expert agreement across multiple statements. Despite the significance, this overall score remains fair to moderate agreement. Can you discuss in the discussion that the concordance is fair to moderate, but the percentage agreement in table 4 is high. There seems to be a discrepancy here.

Reviewer #2: Dear author(s). I appreciate the work you conducted and produced in the submitted article in PLOS ONE. It is a valuable work and needs minor corrections to be complete.

Overall “accept pending revisions”

The idea is great, it has good feasibility and it is worthy to be investigated. It is directional toward finding the best treatment for non-specific neck pain regarding type, dose of exercise. Its value is situated in making an international consensus that will collect the opinions of many experts all over the world.

Abstract

Background: I see it is better to start your abstract with non-specific neck pain and not with neck pain generally.

Conclusion: "Important exercise and dosage variables highlight a complexity of exercise prescription not found in clinical trials". I think this conclusion is not accurate enough. Some papers describe the dosage variables that highlight the complexity of exercise. This is also referred to in the systematic reviews concerned with exercise therapy on CNSNP.

Introduction

- Line 35: What is meant by "effect size" here?

- Line 37: What about manipulating exercise type?

- Line 38- 46: I see that exercise dose is not clarified enough in the introduction section.

- Lines 57-77: Why you neglected the pillar part of exercise and the upper limb?

Aim of the study

- I think the difference between the two purposes you put here. The first purpose is to have expert opinion on exercises and the other is to gain consensus, and I think the consensus will be reached based on the opinion of the expert too. Please make this clear.

Methods

Design

- Line 85: The abbreviation (CREDES) is not clear.

Participants

- Line 95: How many academic and exercise professional expert were recruited to participate in the study?

Round 1

- Line 119: Who has given the overview of exercise to the experts?

- Lines 126- 128: "Previous trial data were collected concurrently during round 1 to allow participants to provide their expert opinion without influence from the literature, reducing experimenter bias.[" What sort of data were provided to the experts and how this data may be influenced if they adapt it from the literature, and how they will give their opinion without returning to the literature and exploring the studied included the exercises used to treat CNSNP?

Round 2

- Line 147: How the thematic analysis of the qualitative data was conducted?

Patient and public involvement

- Line 181: What is meant by public involvement?

- Line 183: What does this abbreviation (GRIPP2-SF) indicate?

Table 1

- How patient can review a systematic review i.e how can he understand the critical appraisal and the conclusions of RCTs? Did they received patient education during having their treatment sessions?

Discussion

- Line 275: Did you investigate the adherence to exercise or the researchers composed a statement regarding this? And if not, please mention why?

- Line 317: What is meant by volume of exercise?

- Line 327: Please revise the font of this title.

Research and clinical implications

- Line 353: What the abbreviation PPI is for?

Conclusion

- Line 364: Exercise position is a new variable mentioned for the first time here.

Reviewer #3: The authors have attempted to establish expert consensus regarding optimal exercise and dosage variables for exercises prescribed to chronic non-specific neck pain patients. However, the methods and procedure section are not clearly explained to recommend the manuscript in its current format for publication.

Introduction:

The authors may consider referring to other studies to develop background for their hypothesis. What’s the difference between objectives 1 and 2?.

Methods:

• It’s not clear how the responses to questionnaires were collected.

• It’s not clear what the authors mean by ‘explain important exercise and dosage variables for each exercise type when treating CNSNP’ - please define each exercise type.

• Previous trial data were collected concurrently during round 1 to allow participants to provide their expert opinion without influence from the literature – How and from where were this data collected?

• Participants were provided with an overview of the systematic review results- provide citation to the SR

• Isn’t it a common practice to rate their agreement with statements and collect feedback on theme/statement generation on round 1 in Delphi studies using a Likert scale?

• What were the criteria to recruit patients in SSG and how were they recruited? Don’t the authors think having the patients with different perceptions of the experts with different educational, professional and cultural backgrounds will affect the outcome? Were the expert members aware of the involvement of patients in the study steering group? Were any statements added based on the recommendation of SSG? If yes, why no description of that has been incorporated in the Delphi procedure section?

• Please clarify – what do the authors mean by ‘Patient representatives suggested the use of “exercise training program” rather than resistance training to collectively describe the exercises described in this study and the systematic review from which this study is based’.

• The point 5- Patient representatives highlighted their importance of being able to self-regulate exercise is not clear;- during which phase (rounds) of Delphi were these themes developed?

• There are so many decisions that seem to be supported by the patients. Considering only 2 patients were present in steering committee, how was it achieved? Can this be called a consensus/ opinion?

• At exactly which part of the Delphi interview, did the patient steering group members felt patients should be made aware that clinicians may not have all the answers and there are gaps in the literature.

6. PLOS authors have the option to publish the peer review history of their article (what does this mean?). If published, this will include your full peer review and any attached files.

Reviewer #1: **Yes: **Francois Maissan

Reviewer #2: No

Reviewer #3: **Yes: **G. Shankar Ganesh

---

## [Author Response · Author response to Decision Letter 0]

27 Apr 2021

We would like to thank all the reviewers and editors who have taken the time in reviewing our manuscript. We have responded to each of the comments in the file "Response to Reviewers" and identified how we have updated the manuscript.

---

## [Decision Letter · Decision Letter 1]

8 Jun 2021

Expert consensus on the important chronic non-specific neck pain motor control and segmental exercise and dosage variables: an international e-Delphi study

PONE-D-21-00717R1

Dear Dr. Price,

We’re pleased to inform you that your manuscript has been judged scientifically suitable for publication and will be formally accepted for publication once it meets all outstanding technical requirements.

Kind regards,

Zubing Mei, MD,PH.D

Academic Editor

PLOS ONE

Additional Editor Comments (optional):

Reviewers' comments:

Reviewer's Responses to Questions

**Comments to the Author**

1. If the authors have adequately addressed your comments raised in a previous round of review and you feel that this manuscript is now acceptable for publication, you may indicate that here to bypass the “Comments to the Author” section, enter your conflict of interest statement in the “Confidential to Editor” section, and submit your "Accept" recommendation.

Reviewer #1: (No Response)

Reviewer #3: All comments have been addressed

2. Is the manuscript technically sound, and do the data support the conclusions?

Reviewer #1: Yes

Reviewer #3: No

3. Has the statistical analysis been performed appropriately and rigorously? 

Reviewer #1: Yes

Reviewer #3: Yes

4. Have the authors made all data underlying the findings in their manuscript fully available?

Reviewer #1: (No Response)

Reviewer #3: Yes

5. Is the manuscript presented in an intelligible fashion and written in standard English?

Reviewer #1: (No Response)

Reviewer #3: Yes

6. Review Comments to the Author

Reviewer #1: The authors have adequately addressed mine comments.

The only thing I wonder is whether there should be a limitation of deep cervical muscles (motor control) and/or the superficial muscles (segmental) so that the pain decreases because these limitations improve or that the pain improves anyway even if there are no limitations of the muscles is present. In other words, is the indication for these exercises pain or muscle impairment? This may also be something to investigate in future research.

I'll leave it up to the authors to add this consideration to the discussion.

Reviewer #3: (No Response)

7. PLOS authors have the option to publish the peer review history of their article (what does this mean?). If published, this will include your full peer review and any attached files.

Reviewer #1: **Yes: **Francois Maissan

Reviewer #3: **Yes: **Shankar Ganesh

---

## [Editor Report · Acceptance letter]

23 Jun 2021

PONE-D-21-00717R1 

Expert consensus on the important chronic non-specific neck pain motor control and segmental exercise and dosage variables: an international e-Delphi study 

Dear Dr. Price:

I'm pleased to inform you that your manuscript has been deemed suitable for publication in PLOS ONE. Congratulations! Your manuscript is now with our production department. 

Kind regards, 

on behalf of

Dr. Zubing Mei 

Academic Editor

PLOS ONE